# Wearables in Swimming for Real-Time Feedback: A Systematic Review

**DOI:** 10.3390/s22103677

**Published:** 2022-05-12

**Authors:** Jorge E. Morais, João P. Oliveira, Tatiana Sampaio, Tiago M. Barbosa

**Affiliations:** 1Department of Sport Sciences, Instituto Politécnico de Bragança, 5300-252 Bragança, Portugal; morais.jorgestrela@gmail.com (J.E.M.); jpco-2001@live.com.pt (J.P.O.); tatiana_sampaio30@hotmail.com (T.S.); 2Research Centre in Sports, Health, and Human Development (CIDESD), University of Beira Interior, 6201-001 Covilhã, Portugal

**Keywords:** wearables, sensors, swimming, monitoring, training

## Abstract

Nowadays, wearables are a must-have tool for athletes and coaches. Wearables can provide real-time feedback to athletes on their athletic performance and other training details as training load, for example. The aim of this study was to systematically review studies that assessed the accuracy of wearables providing real-time feedback in swimming. The Preferred Reporting Items for Systematic Reviews and Meta-Analyses (PRISMA) guidelines were selected to identify relevant studies. After screening, 283 articles were analyzed and 18 related to the assessment of the accuracy of wearables providing real-time feedback in swimming were retained for qualitative synthesis. The quality index was 12.44 ± 2.71 in a range from 0 (lowest quality) to 16 (highest quality). Most articles assessed in-house built (*n* = 15; 83.3%) wearables in front-crawl stroke (*n* = 8; 44.4%), eleven articles (61.1%) analyzed the accuracy of measuring swimming kinematics, eight (44.4%) were placed on the lower back, and seven were placed on the head (38.9%). A limited number of studies analyzed wearables that are commercially available (*n* = 3, 16.7%). Eleven articles (61.1%) reported on the accuracy, measurement error, or consistency. From those eleven, nine (81.8%) noted that wearables are accurate.

## 1. Introduction

In competitive sports, athletes spend a considerable amount of time refining the technique. This enhancement in the technique will likely increase the odds of improving athletic performance at major competitions. Coaches, researchers, and performance analysts usually focus on the athletes’ sports technique because it is highly correlated to performance [1,2]. Hence, detailed and accurate information on the athletic performance and technique in training and competition settings is paramount [3,4]. Additional training details, such as training loads, are other topics of interest. In swimming sports, video analysis is the mainstream procedure to assess the technique [5,6]. However, this assessment procedure is challenging, has a steep learning curve, is very time-consuming, and is potentially disruptive of training sessions and competition [7]. 

During the last two decades, alternatives to video-based assessment techniques have been developed and have gained traction. These include wearables such as accelerometers, gyroscopes, and magnetometers, which are all integrated into one single unit known as “inertial measurement units” (IMUs) [8,9,10]. These sensors can provide a comprehensive set of kinematic variables, such as acceleration (by accelerometers); angular velocity (by gyroscopes); and, rotation, orientation, or heading (by magnetometers) [11]. Several other kinematic and kinetic variables can be derived or extracted from the above-mentioned data. As technology improves, wearables are becoming more affordable, user-friendly, and readily available to end-users (i.e., swimmers, coaches, performance analysts, and applied researchers). Technology developments have led to the compactness of sensors, which can be a great advantage in terms of drag and body placement [10]. The wearable can be placed on a body landmark that: (1) does not increase significantly drag resistance; (2) does not constrain the limbs’ actions; and (3) does not affect the swimmers’ displacement [10]. Incidentally, it is possible to set up several wearables concurrently on the body and increase the number of variables of interest to measure [12]. Swimming wearables can provide real-time details on lap count and stroke count per lap [13,14], swim speed and stroke frequency [15,16], and upper limbs’ asymmetries [17]. They can also detect the stroke being performed [18], as well as estimate the energetics [19] and the thrust produced by the upper limbs [20]. Even though there is considerable interest in technology gathered among end-users, it is unclear how accurate these wearables are.

The aim of this study was to systematically review the current body of knowledge on the accuracy of wearables providing real-time feedback in the sport of swimming. 

## 2. Materials and Methods

### 2.1. Literature Search and Article Selection

As of 10 February 2022, the Web of Science, ScienceDirect, and Scopus databases were searched to identify studies that included any type of wearables in a swimming setting. As an initial search strategy, the title, abstract, and keywords in the text were first identified and read carefully for a first scan and selection of the articles. If one of these fields (title, abstract, or keywords) was not clear on the topic under analysis, the complete article was read and fully reviewed to determine its inclusion or exclusion. The PI(E)CO search strategy used (P—patient, problem or population; I—intervention; E—exposure; C—comparison, control, or comparator; O—outcomes) is presented in Table 1.

The inclusion criteria were: (1) studies written in English; (2) experimental research designs; (3) studies published in a peer-reviewed journal; (4) studies which assess swimming wearables; (5) studies which recruit healthy and able-bodied swimmers; and (6) studies which report on the use of swimming wearables streaming data to provide real-time feedback. Exclusion criteria included: (1) studies not written in English; (2) review papers, conference papers, and books; (3) studies published in non-peer reviewed journals; (4) studies not related to swimming wearables; (5) studies which recruited disabled swimmers or participants with any pathology; and (6) studies not related to the topic in question (e.g., not clearly stated “real-time” or streaming or the sample was not clearly described). Figure 1 depicts the PRISMA flow diagram for identifying, screening, and checking eligibility, which then helps to determine inclusion of the articles. A total of 18 articles [21,22,23,24,25,26,27,28,29,30,31,32,33,34,35,36,37,38] were included in the qualitative synthesis.

### 2.2. Quality of the Articles

The articles selected were screened for quality assessment by an instrument proposed and developed for this scientific field [39,40]. Two independent reviewers read all articles retained for qualitative synthesis and scored the 8 items. Each item was scored 2 points if the answer was “yes”, 1 point if the answer was “partial”, and no points if the answer was “no” [39,40]. Hence, the quality ranges between 0 (lowest quality) and 16 (highest quality). Afterwards, the Cohen’s kappa (K) was computed to assess the agreement between reviewers. It was interpreted as: (1) no agreement if K ≤ 0; (2) none to slight agreement if 0.01 < K ≤ 0.20; (3) fair if 0.21 < K ≤ 0.40; (4) moderate if 0.41 < K ≤ 0.60; (5) substantial if 0.61 < K ≤ 0.80; and (6) almost perfect if 0.81 < K ≤ 1.00.

## 3. Results

The quality index was 12.44 ± 2.71 points. The Cohen’s kappa yielded an almost perfect agreement between reviewers (K = 0.851, *p* < 0.001). Table 2 summarizes the aims of the studies, the participants´ demographics, and the swim strokes. From the 18 articles included for qualitative synthesis, eight (44.4%) [21,23,24,25,30,31,34,38] assessed wearables exclusively in front crawl, six (33.3%) [26,27,29,32,33,35] assessed wearables exclusively in all four swim strokes, two assessed wearables for the tumble turn in front crawl (11.1%) [36,37], one just assessed wearables in breaststroke (5.5%) [22], and another one assessed wearables in butterfly stroke (5.5%) [28]. Overall, 177 swimmers were recruited in all studies (103 males, 62 females, and 12 that the authors failed to note the sex of the participants). Four articles recruited elite-level swimmers (22.2%) [29,32,34,37], two articles recruited international-level participants (11.1%) [33,35], four articles recruited national-level/semi-professional participants (22.2%) [22,28,35,38], one article recruited local-level participants (5.5%) [31], and three articles recruited local/non-expert swimmers (16.7%) [22,25,36]. Conversely, six articles (33.3%) [21,23,24,26,27,30] did not report the swimmers’ competitive level.

Table 3 consolidates the details on the assessed sensor, the anatomical landmark where it was placed, variables analyzed, and the main findings of the studies. Fourteen studied accelerometers (77.8%) [21,22,23,24,25,26,27,29,32,33,35,36,37,38], seven studied microcontroller units (MCUs) (38.9%) [22,25,26,27,28,29,37], six studied IMUs (33.3%) [21,22,23,24,28,31], three studied visual feedback (16.7%) [25,26,27], and just one studied a gyroscope (5.5%) [34].

About the anatomical landmark, eight studies placed the wearable on the lower back (44.4%) [21,22,23,29,34,36,37,38], seven on the head (38.9%) [24,25,26,27,28,33,35], and five on the wrist (27.8%) [21,25,26,27,30] (Table 3). Overall, the articles identified and/or measured variables related to kinematics and kinetics (e.g., duration of the stroke, stroke count, stroke rate, speed) during the arm stroke. Nonetheless, there were three articles that analyzed variables related to the swimmer’s body roll while performing the arm stroke (16.7%) [23,28,34], and two during the tumble turn (11.1%) [36,37]. One article estimated the energy expenditure (5.5%) [30]. Eleven articles (61.1%) [21,22,24,26,27,29,30,31,32,33,35] reported outputs related to accuracy, measurement error, or consistency. Seven papers (38.9%) [23,25,28,34,36,37,38] did not report any information about the reliability and only describe the sensors’ features and specifications. Three articles (16.7%) [30,33,35] selected commercially available wearables and remaining are non-commercial apparatus (i.e., in-house built) (Table 3). From the 11 articles that described accuracy or measurement errors: (1) 9 [21,22,24,26,27,29,31,32,33] (81.8%) reported that the wearable under study could accurately monitor the swim; (2) 1 article [30] (9.1%) conveyed mixed findings depending on the variables measured, and; (3) 1 article [35] (9.1%) noted that the wearable used was not accurate. Detailed information on the accuracy of the outputs can be found in Table 4. 

## 4. Discussion

The aim of this study was to systematically review the current body of knowledge on the accuracy of wearables providing real-time feedback in the sport of swimming. The quality of the research was 12.44 ± 2.71. Most articles assessed in-house built wearables in front crawl, and measured swimming kinematics, placed on the lower back or the head, as well as the accuracy, measurement error, and consistency. The majority of the articles reported the wearables as accurate.

The quality was assessed on a scale from 0 to 16 points and the articles included had an overall score of 12.44 ± 2.71. Thus, the articles in the synthesis are deemed as being of “good” quality, or at least being closer from the upper limit of the scale. Nonetheless, the item where the articles were lacking better scores were related to negative findings. Overall, the articles did not report or elaborate on potential concerns or issues whenever using the analyzed wearables. For sports in general, it was shown that discrepancies can be found in the amount of details given in the studies that used wearable sensors [41]. On the other hand, it was pointed out that wearables can provide accurate information about biomechanical parameters [41]. Besides that, authors reported mostly in-house built wearables. It is possible that they only published their findings after the final solution was fully developed and implemented, thus solving any accuracy issues. Moreover, future research could focus on the independent assessment of wearables developed by third parties.

The articles included in this systematic review recruited swimmers with different demographics, i.e., from the recreational to elite level. From the 12 articles that described the swimmers´ demographic, 8 recruited elite, or international, or national swimmers, or semi-professional-level swimmers. The remaining ones recruited local or non-expert swimmers. Thus, researchers are prone to recruit swimmers with high expertise. This might happen because expert swimmers will be able to deliver the requested task with a good level of proficiency. That said, recruiting participants with different backgrounds is a competitive advantage when assessing the accuracy of sensors catering to a wide range of swimmers.

Near half of the articles retained for analysis (*n* = 8, 44.4%) [21,23,24,25,30,31,34,38] assessed the wearables exclusively in the front-crawl stroke. Indeed, front crawl is the fastest swim stroke [42] and has the largest number of competitive events [43,44,45,46]. Thus, it receives the largest interest among researchers in comparison to the other three competitive strokes (backstroke, breaststroke, and butterfly stroke). The studies aimed to develop or use wearables to monitor the stroke kinematics, including lap count, stroke count, stroke phase detection and its duration (even during breathing), swimming velocity, stroke rate, stroke length, and stroke index. Lap and stroke count are variables that are important for coaches to monitor the training volume, a training load parameter [47,48]. Conversely, literature reports that stroke kinematics (e.g., stroke rate, stroke length, stroke phase durations, etc.) play a major role in swimming performance [1,2,4]. Thus, the devices under study can be used on a daily basis by end-users to monitor the training load (e.g., volume) and swim technique (e.g., stroke rate, stroke length, stroke phase durations, etc.) with the ultimate goal of enhancing performance. Articles that assessed the four swim strokes concurrently [26,27,29,32,33] revealed that breaststroke [22] or butterfly stroke [28] had similar outputs. As for the articles that assessed the tumble turn during the front-crawl stroke, the main focus was to analyze the acceleration in each phase of the turn [36] and to establish the features of the tumble turn performance that could be identified by tri-axis acceleration time-series [37]. In swimming, the turns account for a meaningful contribution to the final race time [46,49]. These days, the turn is a key-phase of the swimming race. Most coaches and swimmers are aware how important is this phase to final race time. Therefore, providing end-users immediate feedback on the turn performance, can be an added value of these wearables.

Incidentally, none of the articles provided data on swimming propulsion. Propulsion is a main determinant of swimming velocity [50]. Moreover, propulsion data can also provide insights on the swimmers’ symmetrical/asymmetrical limb actions [51]. However, research on propulsion-selected equipment suggest that the set-up, collection, extraction, and handling of data are time-consuming and require a high level of expertise. As far as our understanding goes, literature does not provide information on the development or application of wearables that are friendly for end-users. Thus, there is an opportunity to develop wearables than can provide data on the swimmers’ propulsion in a straightforward fashion.

Most articles studied accelerometers. The selection of this type of sensors may be related to the type of task to be performed and assessed, as well as the technology available at a given point in time. For instance, studies on the tumble turn [36,37] used accelerometers and gyroscopes, and the article on the body roll [34] only used a gyroscope. Interestingly, most recent articles developed IMUs, combining the accelerometer, gyroscope, and magnetometer. Integrating these three types of sensors enables a more comprehensive analysis, such as speed, direction, acceleration, specific force, angular rate, and magnetic fields surrounding the device [52]. Moreover, these units provide great self-independence, can work in all environments, and provide good real-time estimation [10,31]. For a seamless experience by the user, the design and development of swim wearable technology faces extra challenges in comparison to available on-land solutions. The fact that the hardware (i.e., the sensor collecting bio-signal) end-users are under water leads to an added constrain. The frequency of limbs´ actions and the ready access to data by streaming it to a third-party device, during or upon collection, should also be considered. Hence, in future pieces of research, authors are encouraged to share details on signal frame rate, feedback delay, wireless communication, among other specifications.

Regarding the anatomical landmarks, where the sensors were set up, the place of choice depends on the variables of interest. To measure swim velocity, wearables were placed on the lower back [22] (i.e., waist level, which can be a good proxy of the swimmer’s center of mass [53]), or the head (which is the landmark used to measures the swimmer’s velocity in race settings [49]). Studies where the wearable was placed on the upper-limb (forearm, wrist, or palm of the hand) aimed to measure the stroke count, stroke rate, and duration, and also helped to identify the stroke phases [21,24,25,26,27,30,31,32]. When the aim was to count the kicking (legs’ downbeat), the wearables were placed on the shanks [24]. Notwithstanding, a major concern around the use IMUs is where they should be placed. Mistakes concerning the placement of IMUs are more challenging than manufacturing variations, environmental conditions, synchronization, or integration drift [54]. Therefore, users must be mindful and careful about the body landmark where the wearable should be placed to yield the expected output.

Only three articles [30,33,35] selected commercially available wearables and the remaining used non-commercial equipment (i.e., in-house built). Hence, wearables are still in early stages of the innovation cycle, are at the development stage, and are not available to end-users. Another key topic concerns the accuracy or measurement error of wearables [41]. Indeed, it was claimed that commercial wearables may not be suitable for professional use due to the low level of accuracy [55]. Thus, the three articles aimed to learn about the accuracy of commercial wearables [30,33,35]. Overall, the study by Lee et al. [30] noted that the error rate of lap count and stroke count at various swimming speeds were within 10% for Apple and about 20% for Garmin. On the other hand, the error rate of estimating energy expenditure was higher for Apple than Garmin. Thus, the authors suggested that Apple and Garmin wearables can accurately measure lap counts and stroke counts, but the energy expenditure estimation is poor at slow or medium speeds. Two studies aimed to assess the validity of the same wearable [33,35]. However, they presented different outputs. Pla et al. [33] reported that the accuracy of the spatial–temporal variables (stroke count, swim speed, stroke rate, stroke length, and stroke index) was high in international open-water swimmers. Conversely, Shell et al. [35] noted that the total swim distance was underestimated by the wearable in comparison to video analysis. Moreover, the authors pointed out that the absolute error in a set of spatial–temporal variables was consistently higher, in comparison to video analysis [35]. Altogether, it seems that commercial wearables should undergo deeper and comprehensive benchmark analyses, in comparison to the gold-standard methods, in order to gain a better insight on its accuracy. The remaining articles [21,22,24,26,27,29,31,32] that used non-commercial wearables and measured accuracy or measurement errors noted that the wearables under study were accurate. Overall, the measured parameters were related to spatial–temporal variables, following identification and determination of the stroke phases. Therefore, based on the data gathered by this systematic review, one can suggest that non-commercial wearables (i.e., in-house-built) seem to report better accuracy than commercial solutions. However, it must be pointed out that seven articles (38.9%) [23,25,28,34,36,37,38] studying non-commercial wearables did not report any information on the reliability. As aforementioned, the main advantage of using such wearables is to spend less time collecting and handling data, thus providing the user with immediate feedback [10,41]. However, based on the findings from this systematic review, it seems that the accuracy of commercially available wearables has room for improvement. Conversely, in-house built systems should move on to other stages of the innovation cycle, thus making them more user-friendly and independently tested.

Overall, it was pointed out that there is potential for wearable technology to be used for long-term monitoring in sports [41], and specifically in swimming. Notwithstanding, wearables can become a tool with pivotal importance in other settings before the user can reach competitive and high-performance levels. This technology can eventually become a mainstream tool in teaching swimming and drowning prevention [56]. Such solutions can be used for tracking and locating the user in water, as well as for drowning detection, and can even deployed as anti-drowning systems. They can also play an important role in health settings; for injury prevention, wearables can provide coaches and athletes with the ability to observe and analyze biomechanical risk factors over a defined exposure time, for example [31,41]. This information can be even more important when delivered in real time. It must be pointed out that one of the “outcomes” of the PI(E)CO strategy was “real-time”. We chose to add this outcome because literature reports the immediate feedback (i.e., in real-time) as one of the main advantages in using wearables [33,41]. However, when performing this systematic review, several articles, e.g., [57,58,59], were not retained because they did not mention this. The main goal of these studies is to understand the validity and accuracy of wearables that provide readily available feedback to coaches and athletes with valuable information that can help them improve their performance. Thus, the competitive advantages of swimming wearables are: (1) they are user-friendly, by decreasing the level of expertise needed to set up the device, as well as collect and handle data; (2) they are less time-consuming, thus providing immediately feed-back; and (3) they can provide data with higher accuracy. Thus, future studies about wearables in swimming or any other sport should clearly mention if the wearable is user-friendly, and if it can provide real-time or immediate feedback as well as improved accuracy.

## 5. Conclusions

The articles retained in this systematic review on the use of wearables in swimming mainly assessed in-house built solutions in the front-crawl stroke; measured swimming kinematics, placed on the lower back or the head; and evaluated the accuracy, measurement error, or consistency. The majority of the articles reported the wearables as accurate.

## Figures and Tables

**Figure 1 sensors-22-03677-f001:**
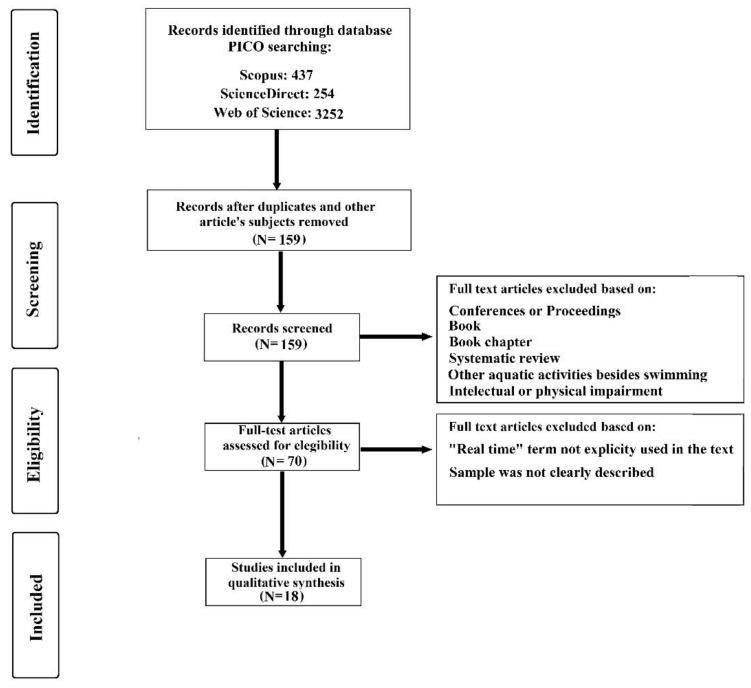
Summary of PRISMA flow for search strategy.

**Table 1 sensors-22-03677-t001:** PI(E)CO (P—patient, problem or population; I—intervention; E—exposure; C—comparison, control, or comparator; O—outcomes) search strategy.

Population	Intervention or Exposure	Comparison (Design)	Outcome
Swimmer *	Validation	Cross-sectional	Performance
	Identification	Longitudinal	Velocity
	Development	Experimental	Stroke frequency
	Measurement	Exploratory	Stroke rate
	Reliability	Descriptive	Stroke count
	Accuracy	Randomized control trial	Heart rate
		Crossover	Energy cost
			Energy expenditure
			Propelling efficiency
			Froude efficiency
			Stroke index
			Critical velocity
			Power
			Mechanical power
			Force
			Propelling force
			Propulsive force
			Body roll
			Speed
			Real time
			Feedback

Note: Asterisks denote truncation to retrieve words with different endings.

**Table 2 sensors-22-03677-t002:** List of the articles selected for qualitative synthesis, including the article aim, the participants’ demographics, and the swim stroke analyzed.

Source	Aim	Participants	Swim Stroke
Dadashi et al. [21]	To present an automatic algorithm to detect main stroke temporal features using IMUs.	Five males and two females: 18.7 ± 5.3 years; 177.4 ± 4.8 cm height; 67.7 ± 5.7 kg body mass.	Front crawl.
Dadashi et al. [22]	To propose a Bayesian framework to estimate breaststroke swimming velocity using a wearable IMU without a priori knowledge of pool length.	Eight well-trained national-level swimmers (six males and two females): 20.7 ± 3.5 years; 181.0 ± 7.9 cm height; 74.1 ± 7.3 kg body mass. Seven recreational swimmers (three males and four females): 14.2 ± 1.0 years; 166.5 ± 12.0 cm height; 55.5 ± 12.2 kg body mass.	Breaststroke.
Engel et al. [23]	To transfer the findings from the video analysis data to the data measured with an IMU.	Six female (14.8 ± 0.9 years) and four male (16.0 ± 0.7 years) swimmers.	Front crawl.
Fantozzi et al. [24]	To develop and validate an easy-to-use tool for stroke-by-stroke evaluation of a swimmer’s integrated timing of stroking, kicking, and breathing.	Twelve male swimmers: 19.1 ± 2.3 years; 76.7 ± 3.7 kg body mass; 179.0 ± 5.2 cm height.	Front crawl.
Hagem et al. [25]	To present a wrist mounted accelerometer and optical wireless communications to display goggles to give real-time feedback to a swimmer during swimming.	One recreational swimmer (male).	Front crawl.
Hagem et al. [26]	To present a smart serial or standard infrared system tested for short-range swimming applications based on stroke rate feedback to maintain the stroke rate at a predetermined swim plan.	One swimmer.	Front crawl.
Hagem et al. [27]	To present a system to provide real-time feedback for swimmers during swimming based on optical wireless communication in the visible spectrum for pacing the swimmer according to the coach instructions.	One swimmer (male).	Front crawl, breaststroke, backstroke, and butterfly.
Jeng [28]	To (1) use open software and hardware to develop a low-cost, portable IMU for swimming movement research; (2) conduct a study in a real-world scenario; and (3) use the developed IMU to analyze the influence of interaction effects among swimmers’ characteristics and breathing patterns on butterfly-stroke swimming efficiency.	Four semi-professional females (average height: 167.2 cm, average body mass: 59.6 kg, average years: 20.8 years). Five semi-professional males (average height: 176.6 cm, average body mass: 74.8 kg, average years: 21.4 years)	Butterfly.
Le Sage et al. [29]	To present a system for automatically capturing the lap count, stroke count, and stroke rate of the swimmer with the results relayed to the coaches in real time.	Four elite swimmers, for each of the four strokes.	Front crawl, breaststroke, backstroke, and butterfly.
Lee et al. [30]	To evaluate the accuracy of the information on lap count, stroke count, and energy expenditure provided by wearable devices (Apple and Garmin) during swimming.	40 males (38.7 ± 11.05 years) and 38 females (39.1 ± 10.67 years) able to swim at various speeds.	Front crawl.
Mangia et al. [31]	To instrumentally validate IMUs and to describe the use of IMUs for multi-body joint kinematics in clinical and sport applications (including swimming).	Six male regional-level swimmers: 26.1 ± 3.4 years, 77.0 ± 10.1 kg of body mass, 182.5 ± 8.8 cm of height.	Front crawl.
Pan et al. [32]	To implement a designed scheme on a water-proof Android smartphone.	Five elite swimmers.	Front crawl, breaststroke, backstroke, and butterfly.
Pla et al. [33]	To evaluate the validity and the reliability of a swimming sensor to assess swimming performance and spatial–temporal variables.	Six international male open-water swimmers (18 ± 3 years).	Front crawl, breaststroke, backstroke, and butterfly.
Rowlands et al. [34]	To present a case study in which inertial sensor time-series data from an elite and sub-elite swimmer were compared using visualization techniques to highlight differences in their action and performance.	Two competent swimmers (one elite and one sub-elite).	Front crawl.
Shell et al. [35]	To independently validate a wearable inertial sensor designed to monitor training and performance metrics in swimmers.	Four males (one national- and three international-level) and six females (one national- and five international-level).	Front crawl, breaststroke, backstroke, and butterfly.
Slawson et al. [36]	To establish characteristics seen in acceleration which pertained to phases of the tumble turn.	One amateur-level triathlete (male).	Front crawl (tumble turn).
Slawson et al. [37]	To establish the features of tumble turn performance that could be identified in tri-axis acceleration traces.	One elite male triathlete (23 years).	Front crawl (tumble turn).
Stamm et al. [38]	To investigate the swim velocity and the acceleration symmetry using IMUs.	Eight national-level male swimmers (aged between 17 and 18 years).	Front crawl.

**Table 3 sensors-22-03677-t003:** Summary of the sensors selected, its specifications (size and weight), the body’s placement, variables analyzed, and the main findings.

Source	Sensor	Specifications	Body’s Placement	Variables	Findings
Dadashi et al. [21]	Three wireless waterproofed IMUs (each with a 3D accelerometer and a 3D gyroscope).	n/a	Both wrists and lower back.	Duration of the arm pull, duration of the arm-push, duration of the recovery, and index of coordination.	It was possible to validate the estimation of front-crawl temporal phases extracted from IMUs. The automatic phase detection method provides timely feedbacks that can be used by sport scientists and coaches. This approach can be modified to fit event detection problems in other types of locomotion.
Dadashi et al. [22])	One waterproofed IMU (with a 3D accelerometer and a 3D gyroscope, a battery, a memory unit, and an MCU).	Dimensions: 50 × 40 × 16 mm. Weight: 36 g.	Lower back.	Velocity.	This study reported an accurate estimation of velocity that relies on the learning of the mapping between compact representation of the inertial signals and target cycle velocity measured by a tethered reference. The method provides immediate feedback on the variability of swimmer’s performance that a coach can use to provide tailored feedback to the swimmer during training.
Engel et al. [23]	One IMU (with a 3D accelerometer and a 3D gyroscope).	n/a	Lower back.	Phases of the arm stroke, roll angle, and angular velocity of the hip.	It was shown that athletes with different skill levels show the same characteristics in their IMU data, which is fundamental for the development of algorithms and for the analysis of the front-crawl swimming stroke, not only considering frequency and number of strokes, but also access to intra-cyclic parameters.
Fantozzi et al. [24]	Five triaxial IMUs equipped with an accelerometer and gyroscope.	n/a	Head, forearms, and shanks.	Wrist entry, head entry, head exit, and leg downbeat.	A protocol for integrated analysis of stroking, kicking, and breathing using inertial sensors in front-crawl swimming was developed and validated in comparison with a video-analysis technique. All investigated accuracy parameters highlighted strong agreement with the gold standard.
Hagem et al. [25]	Acceleration sensors with an MCU and memory for data recording. A system with pre-programmed feedback (audio, tactile, and visual).	n/a	Wrist and head (eyes—goggles).	Phases of the arm stroke, and stroke rate.	The wearable data acquisition, processing, and feedback system was designed, implemented, and tested based on visible light communication in order to give a real-time feedback to a swimmer during swimming. Acceleration data were used for stroke rate determination and optimum transmission time in the stroke cycle.
Hagem et al. [26]	An accelerometer sensor and an MCU at the receiver side that saves the data, decides based on preset conditions, and sends feedback to a display mounted on the goggles.	n/a	Wrist and head (eyes—goggles).	Stroke count, time, stroke rate, stroke length, velocity, and stroke duration.	A short-range optical wireless transceiver was designed and implemented for real-time swimmer feedback applications. The system was based on using an encoder and a decoder with an optical transceiver. The information transmitted was the time duration of one complete stroke, which was updated every stroke and presented to the swimmer using an RGB LED mounted on the goggles.
Hagem et al. [27]	Transmitter that includes an MCU unit with memory. Three-axis accelerometer, power supply, and battery charging circuit are included in the circuit board.	Dimensions: transmitter—35 × 35 mm^2^; receiver—45 × 30 mm^2^.	Wrist and head (eyes—goggles).	Stroke rate.	A second-generation system was designed and implemented. The system was tested with different swim speeds (slow and fast) and different strokes (freestyle, backstroke, breaststroke, and butterfly) to validate the system. These experiments were used to optimize the system and verify that the complete system was viable under different conditions, strokes, and swimmers.
Jeng [28]	Charging and power supply circuits, as well as MCU and IMU data storage circuits.	Dimensions: 53 × 29 × 5 mm. Weight: 15 g (before water case).	Head.	Pitch and roll angles of the butterfly stroke breathing pattern.	To investigate the influence of breathing motions on swimming speed during the butterfly stroke, an IMU was developed. It was showed that significant interaction effects exist between age and average breathing time, significantly influencing swimming efficiency. It also indicated that significant interaction effects exist between gender and the number of breaths taken and between gender and average maximum breathing angle. These results demonstrate the efficacy of the proposed IMU, which could be effectively applied to help coaches and researchers analyze and enhance swimmers’ performance.
Le Sage et al. [29]	An MCU in combination with an RF transceiver, and a tri-axis accelerometer and dual-axis gyroscope.	The packaging (containing all systems) has a combined mass of 110 g. Dimensions: 15 × 9 cm.	Lower back.	Stroke count, stroke rate, and stroke duration.	A novel approach to monitoring free swimming performance with embedded real-time filtering and signal processing was developed. The system exhibits many advantages over current analysis techniques since it offers the opportunity to provide feedback to coaches, performance analysts, and sports scientists in real time, as well as more rapid feedback to swimmers.
Lee et al. [30]	Apple Watch S2 and Garmin Fenix 3HR.	n/a	Wrist.	Lap count, stroke count, energy expenditure.	The error rate of lap counting and stroke counts at various swimming speed were within 10% for Apple and about 20% for Garmin. The criterion measurements and a 95% equivalence test showed that the lap counts and the strokes counts recorded by Apple were within the equivalence zone for all of the exercise intensities measured. Bland–Altman plots showed confidence intervals with relatively small deviations in lap counts and the stroke counts for Apple, and energy expenditure for Garmin. However, the error rate of estimating energy expenditure was higher for Apple than for Garmin. Apple and Garmin wearable watches accurately measure lap counts and stroke counts. However, the accuracy of estimating EE is poor at slow to medium swimming speeds.
Mangia et al. [31]	Seven IMUs.	Dimensions: 48.4 × 36.5 × 13.4 mm. Weight: <22 g.	Thorax, arms, forearms, and hands.	Phases of the arm stroke, arm pull duration, arm push duration, non-propulsive phase duration, and stroke rate.	The use of IMUs can provide several advantages over more expensive and bulky systems, such as (1) simpler and faster setup preparation; (2) less time-consuming processing phase, and (3) the chance to record and analyze a higher number of strokes without limitations imposed by the camera’s volume of acquisition.
Pan et al. [32]	A module that gathers linear acceleration values from the accelerometer every n millisecond.	n/a	Palm of the hand.	Stroke count and stroke identification.	A swimming analysis scheme to count and identify swim strokes using an accelerometer was proposed. The stroke analysis phase identifies stroke styles and counts strokes by finding correlated segments, which can be taken as swimmers’ strokes. It was implemented the designed scheme on a waterproof Android platform. The experiment results indicate that the designed scheme can effectively identify stroke styles and count strokes with more than 87% and 94% accuracies on average, respectively.
Pla et al. [33]	TritonWear (triaxial accelerometer, triaxial gyroscope, and triaxial magnetometer).	n/a	Head.	Lap time, stroke count, velocity, stoke rate, stroke length, and stroke index.	The accuracy of spatial–temporal variables with the use of the TritonWear was high in international open-water swimmers. This device may help coaches to analyze spatial–temporal variables during swim training to determine their relationship with performance. The ease of use, the good accessibility, and the ease in understanding the results of this device allow the coaches to give quick feedbacks and advice to the swimmer during swim training.
Rowlands et al. [34]	Gyroscope.	n/a	Lower back.	Angular velocity of the body roll.	The body roll velocity was captured from the gyroscopic sensor and was used to visualize the time-series data. The visualization techniques that were investigated were time series overlay, phase space portraits (two different methods), ribbon plots, and wavelet scalograms. Obvious differences were observable in all the visualization methods. It was found that all the methods were able to give useful information on the consistency of the stroke cycle. Each of the visualization techniques also showed that the consistency was higher in the elite swimmer than the sub-elite swimmer which was expected. Therefore, these techniques do show merit due to the extra information that can be provided on the swimming action.
Shell et al. [35]	TritonWear (triaxial accelerometer, triaxial gyroscope, and triaxial magnetometer).	n/a	Head.	Distance, stroke count, velocity.	The wearable device investigated in this study does not accurately measure distance, stroke count, and velocity swimming metrics, or detect stroke type. Its use as a training monitoring tool in swimming is limited.
Slawson et al. [36]	Accelerometer.	n/a	Lower back.	Acceleration on the turn approach, rotation, and glide.	Using vision data, it was possible to determine turning phases based on acceleration characteristics. This enabled more complete analysis of turning performance as the approach, rotation, and glide could be individually identified. This is a proactive method to alert users to events, rather than the coach or biomechanist, who have to analyze every output to judge whether something is of significance.
Slawson et al. [37]	MCU, analogue-to-digital converter with an associated sensor, triaxial accelerometer, biaxial gyroscope, digital interface (to enable the connection of additional memory), crystal oscillator, radio components, and power.	Dimensions: 90 × 40 mm.	Lower back.	Acceleration on the turn approach, rotation, and glide.	Using the visual data, it was possible to understand the turning phases in acceleration space. Rotation information, last stroke to wall, and first stroke after the turn timing could be automatically distinguished from the captured acceleration data. Turning information, automatically cropped from the raw data stream within these stroke time limits, enabled a more complete analysis of turning performance than what has been previously possible, as the approach, rotation, and glide features could be individually identified and quantified.
Stamm et al. [38]	Triaxial accelerometer, triaxial gyroscope, and radio frequency capabilities.	n/a	Lower back.	Stroke rate, stroke duration, and velocity.	Considering the small and light weight of the used sensor, it can be nearly used during every swimming session. This offers the opportunity to all athletes and coaches to record as many swimming sessions as they want without complex and bulky equipment. It allows arm symmetry investigations in different ways (stroke rate, acceleration, and velocity) and offers the possibility of keeping track of training progress or injury recovery. Furthermore, it brings up the opportunity to identify symmetry problems in swimming styles and helps to adjust the swimmers´ style if necessary.

3D—three dimensions; IMU—inertial measurement unit; LED—light-emitting diode; MCU—microcontroller unit; RF—radio frequency; RGB—red, green, blue; n/a—not applicable or not disclosed.

**Table 4 sensors-22-03677-t004:** Summary of the accuracy of the outputs.

Source	Accuracy of the Outputs
Dadashi et al. [21]	The mean difference between the wearable and video-based systems (accuracy) was always lower than 0.8 frames in detecting the start of each phase. The standard deviation of the difference (precision) was always lower than 2.7 frames (which is on average equivalent to 5.1% of the cycle duration). Bland–Altman analysis also yielded a good agreement, where more than 80% of the plots were within the 95CI.
Dadashi et al. [22]	It was showed that the linear least squares method has poor generalization characteristic for the study’s purpose; however, the Gaussian process and the Bayesian approach are equally accurate methods. The lower computational cost of the Bayesian model suggests it as the preferable method because computational resource is one of the main concerns in standalone wearable platforms.
Engel et al. [23]	n/a
Fantozzi et al. [24]	The respective average RMSEs and 90th percentile of absolute errors for IMU vs. video-based analysis were 0.1 and 1.5% for wrist entry, 0.7 and 8.0% for head exit, 0.1 and 2.2% for head entry, and 0.1 and 1.8% for leg downbeat. Linear regression between methods revealed a nearly perfect agreement (r > 0.90 for wrist entry, head exit, head entry, and leg down beat). Bland–Altman showed that more than 80% of the plots were within the 95CI.
Hagem et al. [25]	n/a
Hagem et al. [26]	For front-crawl stroke, the results showed that LED4 (center of the wrist) and LED6 (outside top edge of the wrist) contributed to 55% and 95% of link availability, respectively. For the remaining swim strokes, the results in water showed that breaststroke had generally a high link availability.
Hagem et al. [27]	The results from the swim style test experiment showed that breaststroke had the highest received data with 60.3%, and 54.3%, 50.5%, and 45.25% for the freestyle, butterfly, and backstroke, respectively. The air test showed that the system is error-free for 70 cm with the top-view LEDs and 35 cm with the side view mounted on the wrist strap. The swim test for different strokes showed that breaststroke was the best with 60.3% error-free received data compared to other swim strokes.
Jeng [28]	n/a
Le Sage et al. [29]	The difference in the real-time processing in butterfly, in comparison to the manual digitization, was determined to be on average 0.07 cycles∙min^−1^. The mean difference was calculated as 0.34 cycles∙min^−1^ for backstroke, 0.25 cycles∙min^−1^ for breaststroke, and 0.10 cycles∙min^−1^ for freestyle. The standard deviation for all four strokes fell under 2 cycles∙min^−1^. Butterworth filter with selected cutoff frequencies can be utilized to minimize the noise components of the signal and allow simple feature extraction algorithms to provide an accurate determination of the stroke characteristics of the swimmer.
Lee et al. [30]	For lap counts and stroke counts, the mean absolute percentage error of Apple was within 10% (lap counts: 0.5 to 6.1%, stroke counts: 6.2 to 9.3%); however, Gamin was about 21% (lap counts: 0 to 20.6%, stroke counts: 6.8 to 17.6%). Both devices showed higher error rates when the speed was slower. Apple overestimated energy expenditure at all speeds. The mean absolute percentage error of speeds between 0.4 m∙s^−1^ and 1.0 m∙s^−1^ were vigorous (32.70% to 151.66%), but the mean absolute percentage error (17.93%) became lower at the speed of 1.2 m∙s^−1^. Garmin showed a mean absolute percentage error of 17.9% to 32.7%. Both wearables showed a tendency to gradually decrease the mean absolute percentage error as the intensity of exercise increases.
Mangia et al. [31]	The instrumental tests quantified a dynamic orientation estimation accuracy of about 6°. This accuracy was lower than that obtained with the standard stereophotogrammetric system. Nonetheless, it was considered enough to provide useful information about swimming.
Pan et al. [32]	The designed scheme can effectively identify stroke styles and count strokes with more than 87% and 94% accuracy on average, respectively.
Pla et al. [33]	The mean absolute percentage error in the 800 m freestyle showed a high level of accuracy for lap time, stroke count, swim speed, stroke rate, stroke length, and stroke index (with all variables under 5% of mean absolute percentage error), in comparison to video recordings (0.9, 3.3, 0.7, 2.9, 4.5 and 4.8%, respectively). In the medley test (i.e., four swim strokes), sensors showed a high level of accuracy for lap time with a mean absolute percentage error of 2.2, 3.2, 3.4, and 4.1% in butterfly, backstroke, breaststroke, and freestyle, respectively. No stroke count error was found in the butterfly stroke. The accuracy was 2.4% in breaststroke, 4.9% in freestyle, and 7.1% in backstroke. The mean absolute percentage error of speed and stroke rate were under 5% in all swim strokes.
Rowlands et al. [34]	n/a
Shell et al. [35]	The total distance swam was underestimated by the device in comparison to video analysis. When benchmarked to video analysis, the absolute error was consistently higher for total and mean stroke count, as well as the mean velocity. Across all sessions, the device incorrectly detected the total time spent in backstroke, breaststroke, butterfly, and freestyle as 51%. The device did not detect the time spent in drills.
Slawson et al. [36]	n/a
Slawson et al. [37]	n/a
Stamm et al. [38]	n/a

RMSE—root mean square error; IMU—inertial measurement unit; 95CI—95% confidence intervals; LED—light emitting diode; n/a—not applicable or not clearly disclosed.

## Data Availability

Not applicable.

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
