# Peer review of "Wearables in Swimming for Real-Time Feedback: A Systematic Review"

_sensors, 2022, doi:10.3390/s22103677_

Round 1
Reviewer 1 Report
Overall, excellent paper that met their stated goal. I appreciated the comparison of commercially available vs in-house built devices and how this represents the early developmental phase of the technology. I enjoyed the authors discussion on the importance of "real-time" feedback and it's application in ensuring the health and safety of athletes during training and competition. I challenge the authors to consider, whether within this paper or the next, to also consider the role of how these technologies overlap with the world of drowning prevention wearables. There is an opportunity for cross-pollination of the two worlds that can improve the technology in both.
Author Response
Kindly refer to attached file with point-by-point replies.

Reviewer 2 Report
The aim of this study was to review the accuracy of wearables providing real-time feedback in swimming. Some interesting information was provided, such as:1) Most (83.3%) of qualified articles assessed in-house built wearables, not commercially available wearables; and 2) Only 9 qualified articles reported that wearables are accurate, all are in-house built. These indicated that it is very challenged to develop an accurate sensor that provides real-time feedback in swimming. Therefore, it would be very valuable to review the method and technology of these wearables. Several concerns need to be addressed:
From line 137 to 141, the authors reported what sensors were accurate and what sensors were not accurate, but did not provide accuracy values. Table 3 summarized the main findings of the sensors selected, including some accuracy information. Since the wearable’s accuracy is one of the main topics of this study, it would be helpful to provide a dedicated summary table that compared the accuracy values (or measurement error values) of apparatus reported by the studies reviewed.
This study focused on sensors that can provide real-time feedback. It would be helpful to compare the signal frame rate and feedback delay of the apparatus because low frame rate would limit the performance of the apparatus in detecting fast movements of swimmers, and the apparatus with low frame rate and long feedback delay might lack of capability of providing immediate feedback to athletes and coaches. Besides, wireless communication distance of the apparatus might be also valuable to study, since wireless communication under water is more difficult than in air, and reliable communication is important to real-time feedback especially for coaches who are not near athletes.
Finally, it would be nice to include the weight and dimension of sensors in this study, e.g., in Table 3; these are key specifications for wearables.
Author Response

(The authors gave the same response as above.)
